# The Balanced Structure of Environmental Identity

**Coral M. Bruni** [1,*] **, P. Wesley Schultz** [2,*] **and Anna Woodcock** [2]

1   Division of Behavioral and Organizational Science, Department of Psychology, Claremont Graduate University, Claremont, CA 91711, USA

2   Department of Psychology, California State University, San Marcos, CA 92096, USA; woodcock@csusm.edu

\*   Correspondence: coral.bruni@cgu.edu (C.M.B.); wschultz@csusm.edu (P.W.S.)

**Abstract:** Connectedness with nature refers to an individual's beliefs about their relationship with the natural environment. The current paper integrates connectedness with nature into a broader framework of balanced identity theory as a form of self-concept, and presents new data showing that individuals tend toward balanced-congruity and hold cognitive configurations that balance self-concept, environmental attitudes, and self-esteem. In essence, when an individual scores highly on one of these constructs, it is likely that they will score highly on the other two constructs. Two hundred and seventy-six undergraduate students completed explicit and implicit measures of connectedness with nature, attitudes toward nature, and self-esteem. The balanced-congruity principle was supported with implicit measures (e.g., Implicit Association Test), but not explicitly with self-report measures. Results suggest that attitudes toward nature, connectedness with nature, and self-esteem form a balanced triadic structure of implicit environmental identity. The findings extend our understanding of connectedness with nature, by integrating it into a broader framework that links connectedness, attitudes, and self-esteem as a triadic form of environmental identity. This finding has important implications for practitioners interested in fostering environmental identities and promoting sustainability.

**Keywords:** balanced identity theory; environmental identity; connectedness with nature

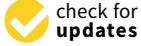



Psychological research on sustainability has focused on understanding the thoughts, feelings, and behaviors of individuals who are associated with environmental protection. Some of the early work in this area examined attitudes and the evaluative judgements that individuals hold about the environment and about conservation [1,2]. Subsequent research extended these findings to explore values and the role of personal and cultural values as underpinnings of environmental attitudes and behaviors [3–5]. More contemporary research has examined the construct of connectedness with nature and the role of personal experiences in cultivating both attitudes and values [6–8]. In the current paper, we draw on social psychological work on cognitive consistency and balanced identity theory to offer an integrated theoretical framework for understanding the relationships between self, nature, and affect.

## 1. Connectedness with Nature

*Connectedness with nature* is a relatively new research area within environmental psychology. In our previous work, we have described connectedness as part of a person's self-concept [8–10]. In this tradition, it refers to an individual's belief about the extent to which they are part of the natural environment. This definition draws on social psychological research regarding the cognitive representation of self, and self-schemas [11–13]. While this line of work continues, the term has since been used to encompass the emotional connection that a person forms with nature [6,14]. While these two research approaches to connectedness have taken different foci, research has shown positive correlations between the cognitive Inclusion of Nature in Self Scale (INS) and the more emotion-based Connectedness to Nature Scale [15,16].

The inclusion model of environmental concern proposes that connectedness with nature is associated with the types of concerns that individuals hold about environmental issues [8,16]. Individuals with high levels of connectedness tend to develop broader, and more self-transcendent concerns about the harmful consequences of environmental problems. These *biospheric* concerns focus on the inherent value of nature, irrespective of any direct impact on self. In contrast, individuals with low levels of connectedness—who see self as separate from nature—tend to hold more self-enhancing concerns about environmental issues. These *egoistic* concerns focus on threats to one's health, lifestyle, loved ones, and prosperity. Importantly, individuals with varying levels of connectedness can express concern about environmental issues, but for fundamentally different reasons [17].

Research drawing on the inclusion model has explored a range of research topics [18]. For example, research has shown positive correlations between connectedness and biospheric environmental concerns [19–21]. Studies in environmental education have examined the types of experiences and educational activities that can increase connectedness with nature [9,22–24], and the role of self-expansion activities such as mindfulness, meditation, and perspective taking in increasing concerns for environmental issues [10,25,26]. Finally, studies have examined the impact of nature experiences, such as living in close proximity to green and blue spaces, in fostering increased levels of connectedness with nature [9,27].

## 2. Environmental Attitudes

Environmental attitudes are the "collection of beliefs, affect, and behavioral intentions a person holds regarding environmentally related activities or issues" [28] (p. 458). Environmental attitudes were defined as "a psychological tendency expressed by evaluating the natural environment with some degree of favour or disfavour" [29] (p. 80). Environmental attitudes, then, are multidimensional constructs, that are organized in a hierarchical fashion from generalized environmental attitudes to attitudes regarding preservation and utilization [30]. In summarizing the research findings regarding environmental attitudes, those identified as holding environmental attitudes were:

> Older, female and members of an environmental organization, who attribute greater importance to self-transcendence, biospheric and altruistic values, who conserve the environment by performing ecological behaviors, who feel connected with nature and concerned about threats from environmental problems, and who support sustainability principles [30] (p. ii).

Research on environmental attitudes and connectedness with nature has found a positive relationship between connectedness with nature and environmental attitudes [6,17,20,31], with those who hold stronger environmental attitudes also having higher connectedness with nature. This relationship between environmental attitudes and connectedness with nature also tends to manifest in more sustainable practices [32]. More so, the relationship between experiences in nature and pro-environmental behavior has been shown to be mediated by environmental attitudes [7,33,34]. Importantly, environmental attitudes themselves have been linked to pro-environmental behaviors [35–39]. This direct link between environmental attitudes and pro-environmental behavior is in line with the Theory of Planned Behavior [40] and the Value-Belief-Norm Theory [41,42].

### *Environmental Identity*

While research into connectedness with nature and environmental attitudes continues to grow, it's helpful to consider connectedness and attitudes within a broad framework of identity. As we have described it, connectedness with nature is the strength of the cognitive association that an individual holds about themselves and the natural environment [20,21]. But this notion of connectedness with nature can serve as part of a person's identity [43]. In general, identity refers to the groups and roles to which a person belongs [44]. These identities are typically social in nature—for example, nationality, gender, or political affiliation—but these identities can extend to membership in environmental organizations,

and even to nature itself. In a recent study of the role of social networking on environmental identity, environmental identities were found to be strengthened by social networking sites which provided a context in which users could both reinforce their beliefs and values, and also mimic the behavior of other users [45]. The authors suggest that identity is a "process which is formed by socio-cultural requirements" [45] (p. 8). Environmental identity has been defined as "one part of the way in which people form their self-concept: a sense of connection to some part of the nonhuman natural environment" [46] (p. 46).

### 3. Balanced Identity Theory

We conceptualize connectedness with nature as one aspect of a person's environmental identity. We have defined connectedness as cognitive—as the strength of an individual's self-nature associations. But identity also includes an affective element, of likes and dislikes. Balanced identity theory [47] offers a unified framework for linking these cognitive and affective dimensions of environmental identity.

Balanced identity theory applies the principles of cognitive-affective consistency theories (e.g., balance theory; cognitive dissonance theory, congruity theory) [48–50] to explain the formation and maintenance of an individual's identity. This unified theory defines identity as the strength of association between self, groups, attributes, and valence. Balance, as suggested through cognitive-affective consistency theories, results when the strength of relationship between two concepts (e.g., self and gender) is commensurate with a third concept (e.g., a positive valence, such as good; or an attribute such as "nurturing"). For instance, in a balanced identity configuration, a person might hold beliefs that "I am female, I am good, and females are good". In an unbalanced configuration, a person might hold beliefs that "I am female, females are nurturing, but I am not nurturing".

Balanced identity theory focuses on the formation and maintenance of a person's identities [47]. In this theory, identities start with the concept of self and relate the self to other object concepts (e.g., groups, attributes, and valence). These self and other attributes are then integrated to form a triad of cognitive associations, with each association embedded within a cognitive network. Concepts within the triadic identity relationships are considered "nodes" and include the self, a social category, and an attribute or valence (positive or negative). An example applying balanced identity theory to an environmental identity can be seen in Figure 1. In this figure, self is related to concepts of nature and valence to form an environmental identity. For instance, I am part of nature (connectedness with nature), I am good (self-esteem), nature is good (attitudes toward nature). The strength of the association between the concepts is expected to vary. Thus, the triadic design examines identity associations (self in association with a social category and an attribute/valence) and can reveal consistencies within the triad of associations.

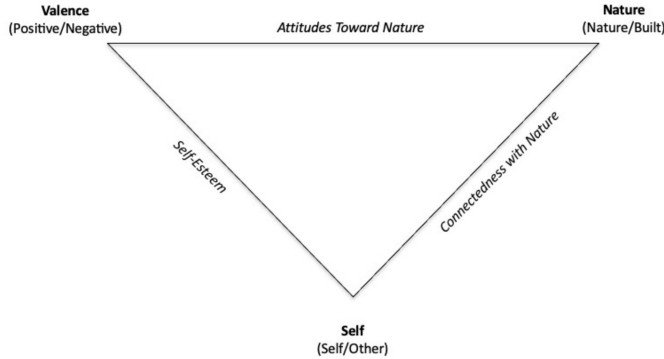

**Figure 1.** Triadic structure of environmental identity.

Balanced identity theory draws on the principles of balance theory [48] and congruity theory [50]. Balanced identity theory proposes that when two of the nodes in the triad are

linked to a common third node, they share a first-order link. The *balance-congruity principle* predicts that when two nodes share a common first-order link, the association between them should strengthen. As an example, consider the three nodes shown in Figure 1: self, nature, and valence. In this example, self and nature both share a first-order link with valence. According to the balance-congruity principle, this shared first-order link exerts pressure on the cognitive system toward equal strengths of association. As a result, a person with a strong positive self-nature association, should also show strong positive self-valence associations (self-esteem) and strong positive nature-valence associations (attitudes). Note that in the triadic structure, each of the three nodes (self, nature, valence) has a set of shared first-order links.

The balanced identity theory relationships can be extended to environmental identity by incorporating the concept of connectedness with nature, attitudes towards nature, and self-esteem [47]. There is a growing body of psychological research examining an individual's beliefs about the degree to which they are part of the natural environment [6,8,17,20,21,51–58]. Implicitly, connectedness with nature has been studied using the IAT-Nature [18,19]. In the IAT-Nature, two target discriminate categories (*nature* versus *built*) and two attribute categories (self versus other) were used to measure the associated strength of the relationship between self and nature. The "nature" category is represented by the words flower, tree, mountain, waterfall, and butterfly and is thought to represent things that occur naturally in our environments. The "built" category is represented by the words church, car, truck, chair, and boat and represents things that do not occur naturally in our environments (e.g., artificial). The "self" category is represented either by pronouns representing the self (e.g., me, myself, I) or by the participants name. The "other" category is represented by pronouns representing others (e.g., other, they, them) or names of other people.

The cumulative results in this area of research have yielded several clear findings [20,21,51,52]. First, there is a consistent IAT effect across studies with most participants showing stronger self-nature associations than self-built associations. Second, these self-nature associations show a moderate level of test-retest reliability, even across a 4-week period ($r = 0.49$). Finally, in most cases, the IAT scores tend to correlate in meaningful ways with explicit measures of environmental attitudes. These effects have been seen across various settings (e.g., urban settings, zoos, universities) and across various populations (e.g., general public, children, university students, environmental activists).

## 4. Hypotheses

The purpose of the current study was to apply balanced identity theory to environmental identity. The reported study was designed to examine the relationships between connectedness with nature, attitudes toward nature, and self-esteem, utilizing the balance-congruity principle from balanced identity theory. Balanced identity theory has typically relied on evidence from implicit measures of social cognition, rather than self-report measures [59] although newer data suggests that balance may also emerge in explicit measures [60]. In the current study, the relationships were examined using both implicit and explicit measures. The structure of balanced identity theory is well suited to testing these relationships (see Figure 1).

Based on the balanced-congruity principle, we hypothesized that attitudes toward nature would be a function of the strengths of connectedness with nature and self-esteem (self/attribute associations). That is, if there is a positive association between self and nature, and between self and valence, then the positive association between nature and valence should follow. This suggests an expected shared first order link between attitudes toward nature and both connectedness with nature and self-esteem. Similar patterns were hypothesized for attitudes toward nature, which should be explained be a function of connectedness and self-esteem. And finally, self-esteem should be a function of connectedness and attitudes. We further hypothesized that the expected patterns of identity balance would be found using implicit measures, but not explicit.

A 4-test process for assessing the balanced-congruity principle was proposed [47]. The analyses are conducted separately for each of the three nodes in the triadic structure. In Test-a, the strength of association between two nodes is predicted in a hierarchical regression model using the product of the other two associations (the multiplicative term). On this first step of the regression model, the multiple *R* should be statistically significant, and the regression coefficient for the multiplicative term should be positive. After testing the multiplicative term, the two individual predictors are added to the regression equation. Test-b predicts that when the individual predictors are added, the original multiplicative term should remain positive. Test-c predicts that the addition of the two predictor variables will not result in a statistically significant increase in the multiple *R* value. Finally, Test-d predicts that neither of the individual predictors should be statistically significant. This analytic sequence is tested three times, once with each leg of the triadic model as the outcome variable. In essence, this analytic procedure tests the extent to which each leg in the triadic model is "balanced" with the other two legs.

## 5. Methods

### 5.1. Participants

Participants in the study were 276 undergraduate students from California (58 male, and 218 female), ranging in age from 18 to 50 (*M* = 21.44, *SD* = 4.56).

### 5.2. Measures

To examine the balanced structure of environmental identity, we measured connectedness with nature, attitudes toward nature, and self-esteem, at both the implicit and explicit levels. Participants completed an online questionnaire to measure explicit self-esteem, attitudes toward nature, and connectedness with nature. Using an online version of the Implicit Association Test (IAT), participants also completed three IATs to measure the same three constructs (see Table 1 for details on the categories for each of the three IATs). In addition, Table 2 highlights the constructs/stimuli used in the explicit Likert questions and the implicit IATs.

**Table 1.** Three IATs used to measure concept associations related to balanced environmental identity.

| IAT | Associated Concepts | | |
|---|---|---|---|
| Connectedness with Nature | Self/Other | ⟷ | Nature/Built |
| Self-esteem | Self/Other | ⟷ | Valence |
| Attitudes towards Nature | Nature/Built | ⟷ | Valence |

Note: Valence refers a positive or negative affect, such as good/bad or positive/negative.

**Table 2.** Constructs/stimuli used to measure concept associations related to balanced environmental identity, both explicitly and implicitly.

| Categories | Stimuli |
|---|---|
| Nature | Tree, Mountain, Butterfly, Flower, Waterfall |
| Built | Boat, Car, Chair, Truck, Church |
| Self | The participant's name |
| Other | Random list of other names |
| Positive | Joy, Warmth, Gold, Happy, Smile |
| Negative | Gloom, Agony, Pain, Stink, Filth |

Nature, built, self, and other stimuli were taken from Bruni and Schultz (2010). Positive and negative stimuli were taken from Greenwald et al. (2002).

### 5.2.1. Explicit Measures

The balanced identity design requires each of the measures (explicit and implicit) to be scored on a scale that has a rational zero point [47]. The location of the zero point is important for several reasons. First, according to Aiken and West (1991 as cited by Greenwald et al., 2002) [47] results from hierarchical regression analyses need to be scaled

for clarity in understanding the results. This scaling procedure holds that a score of zero indicates no strength of the association between the variables. If scaling is not met, the statistical analyses in the second step of the analyses may be inflated and produce spurious statistically significant results. Additionally, this procedure allows for comparability between the explicit and implicit measures, as it creates a form of standardized difference score, similar to that used in the implicit measures (see the implicit measure section for a brief discussion on the D-score). Explicit measures were constructed with these guidelines in mind, and additional scoring procedures were implemented when measures did not meet this prescribed requirement [47].

Three explicit measures were constructed for self-esteem, attitudes toward nature, and connectedness with nature.

**Self-esteem.**

Explicit self-esteem (association of self with positive attribute) was measured following procedures outlined in Greenwald et al. (2002) [47]. Two measures were used to create an overall self-esteem score: a feeling thermometer, and a Likert scale.

*Thermometer*. Participants rated the extent to which they find themselves and other people to be unfavorable or favorable using a thermometer scale ranging from 0 (unfavorable) on the left, 5 (neutral) in the middle, and 10 (favorable) on the right. Using two items, participants rated themselves and other people on the 0-10 scale. This measure was scored using a difference score in which "yourself" was subtracted from "other people" and the resulting score was divided by its standard deviation.

*Likert Scale*. Participants rated the extent to which ten positive and negative words were characteristic of themselves and other people, using a 7-point scale from 1 (Not at all Characteristic) to 7 (Extremely Characteristic). There were five positive items, and five negative, and each was rated for self and then again for other people. The positive and negative stimuli are shown in Table 2. This measure was scored by subtracting the average score of the five unpleasant items from the average score of the five pleasant items. A difference score was created to mimic the results of an IAT self-esteem D-score. To start, the positive, negative, yourself, and other people items were averaged to create compatible and incompatible components, similar to the IAT (self/negative, self/positive, other people/positive, and other people/negative). The self/negative and other people/positive items were then averaged to create an incompatible score. The self/positive and other people/negative items were averaged to create a compatible score. Both the compatible and incompatible scores were divided by their respective standard deviations. Finally, the incompatible score was subtracted from the compatible score, thereby creating a standardized difference score similar to the IAT D-score.

To create an overall self-esteem score, the average of the measures (Thermometer and Likert) was calculated. This procedure preserves the desired location of the zero point.

**Connectedness with Nature**. Explicit connectedness with nature (association of self with nature) was measured used a similar procedure to that reported above for self-esteem:

*Thermometer*. The thermometer measure for connectedness with nature was similar to that used for self-esteem. Participants rated the extent to which they viewed themselves and other people to be more similar to built environments or nature, using a thermometer scale with anchors of 0 (built) on the left, 5 (neutral) in the middle, and 10 (nature) on the right. Using two items, participants rated "self" and "other people" on a scale from built to nature. To create a connectedness with nature thermometer score, a difference score was created by subtracting the average other score from the average self score. Finally, the difference score was divided by its standard deviation.

*Likert Scale*. Participants rated the extent to which each of the constructs of nature and built were characteristic of themselves and other people on a 7-point scale with anchors from 1 (Not at all Characteristic) to 7 (Extremely Characteristic). Using this Likert scale, a total of 20 items were rated, with five positive and five negative items each scored for self and for other people. The Nature and Built stimuli are shown in Table 2.

A difference score was created to mimic the results of an IAT connectedness with nature D-score. To start, the built, nature, yourself, and other people items were averaged to create compatible and incompatible components, similar to the IAT (built/yourself, built/other people, nature/yourself, and nature/other people). The average of the built/yourself and nature/other people items were then averaged to create an incompatible score. The built/other people and nature/yourself items were averaged to create a compatible score. Both the compatible and incompatible scores were divided by their respective standard deviations. Finally, the incompatible score was subtracted from the compatible score, thereby creating a difference score comparable to the IAT D-score.

Similar to the overall self-esteem score, an overall connectedness with nature score was created using by averaging the scores from the thermometer and Likert scales.

**Attitudes Toward Nature**. Explicit measures of attitudes toward nature (association of nature with positive valence) were based on Greenwald et al.'s (2002) explicit ratings of gender attitudes, and a similar procedure to that reported above [47].

*Thermometer*. The thermometer measure for attitudes toward nature was similar to that used for self-esteem, with positive and negative ratings of both nature and built stimuli. Participants rated natural and built stimuli on a thermometer scale with anchors at 0 (negative) on the left, 5 (neutral) in the middle, and 10 (positive) on the right. To create an attitudes-toward-nature thermometer score, the average was computed for the built stimuli and nature stimuli. Next, a difference score was created by subtracting the average nature score from the average built score. Finally, the difference score is divided by its standard deviation.

*Likert*. Participants rated the extent to which each of the stimuli selected to represented nature and built on a 7-point scale with anchors from 1 (Not at all Characteristic) to 7 (Extremely Characteristic). Positive and negative stimuli, along with the built and natural stimuli, are shown in Table 2.

Similar to the explicit connectedness with nature Likert score, a difference score was created to mimic the results of an IAT attitudes toward nature D-score. To start, the built, nature, positive, and negative items were averaged to create compatible and incompatible components, similar to the IAT (built/positive, built/negative, nature/positive, and nature/negative). The average of the built/positive and nature/negative items were then averaged to create an incompatible score. The built/negative and nature/positive items were averaged to create a compatible score. Both the compatible and incompatible scores were divided by their respective standard deviations. Finally, the incompatible scores were subtracted from the compatible scores, thereby creating a difference metric comparable to the IAT D-score.

An overall attitude towards nature score was created by taking the average of both the attitude toward nature thermometer and Likert scores.

### 5.2.2. Implicit Measures

Implicit associations were measured using an online version of the Implicit Association Test (IAT). The three IATs were based on the traditional procedure outlined by Greenwald et al. (1998, 2003), and Greenwald and Farnham, 2000 [61–63]. Three versions of the IAT were used: Self-Esteem IAT (Me = Good) [63], IAT-Nature (Me = Nature) [20,21], and a newly-developed Attitudes-Toward-Nature IAT (Nature = Good).

The IATs were administered using the traditional 7-block sequence. There were four categories of words used per IAT (see Table 1). The blocks were presented as follows:

Block 1: Category Pair 1 Practice
Block 2: Category Pair 2 Practice
Block 3: Congruent Pair Practice
Block 4: Congruent Pair Test
Block 5: Category Pair 2 Practice Reversed
Block 6: Incongruent Pair Practice
Block 7: Incongruent Pair Test

These three IATs were given in a counterbalanced order to reduce potential order effects. Stimuli used within each category are presented in Table 2. The words were presented in random order within each of the blocks. To account for outliers and errors, each response time less than 300 milliseconds (ms) or greater than 3000 ms was excluded from further analysis. The D-score [62] was calculated by comparing reaction times, in milliseconds, of compatible trials to incompatible trials. Using the scoring procedure, D-scores typically range from −2 to +2, with a score of 0 representing no preference. For interpretation, scores of 0.5 are considered medium in size, and scores above 0.8 are consider large.

Following the procedure outlined by Bruni and Schultz (2010) [52], the reliability of the three IATs in the current study were calculated by correlating the two subscales (D1 and D2) for each of the three IATs. Across all three IATs, D1 and D2 were significantly correlated (attitude toward nature IAT: $r = 0.45$, $p < 0.001$; connectedness with nature IAT: $r = 0.31$, $p < 0.001$; self-esteem IAT: $r = 0.32$, $p < 0.001$).

### 5.3. Procedure

Each participant was tested individually. After providing informed consent, participants were asked to complete an online questionnaire, followed by three counterbalanced IATs. Following the completion of the IATs, participants were debriefed.

## 6. Results

Prior to running the analyses, responses were screened for error rates across the three IATs. Due to high error rates on one or more of the three IATs they completed (greater than 30% incorrect), 22 participants were screened from further analyses. One additional participant was eliminated due to incomplete data on explicit measures. The analyses were conducted on a final sample size of 253.

### 6.1. Descriptive Statistics

Following the screening of the data, the means, standard deviations, and overall patterns within the data were examined to show how the triadic relationships of concepts. Descriptive statistics for the primary measures are shown in Table 3. The explicit and implicit triadic relationships were both examined. Implicitly, participants showed positive attitudes toward nature ($M = 0.80$, $SD = 0.38$), and moderate connectedness with nature ($M = 0.48$, $SD = 0.34$), and self-esteem ($M = 0.58$, $SD = 0.34$). In looking at the distribution of the D-score for the attitude toward nature IAT, 98% of participants associated nature more strongly with positive than with negative stimuli, with scores ranging from −0.24 to 1.93. Participants also associated the self more strongly with nature than with built environments (93%), with scores ranging from −0.42 to 1.46. Finally, 97% of participants associated the self more strongly with positive than with negative stimuli, with scores ranging from −0.29 to 1.42.

**Table 3.** Means and standard deviations for the primary measures.

| Measure | *M* | *SD* | Range |
|---|---|---|---|
| **Implicit Association Tests** | | | |
| IAT-Self Esteem | 0.58 | 0.34 | −0.29 to 1.42 |
| IAT-Attitudes toward Nature | 0.80 | 0.38 | −0.24 to 1.93 |
| IAT-Connectedness with Nature | 0.48 | 0.34 | −0.42 to 1.46 |
| **Explicit Measures** | | | |
| Attitudes toward Nature | 1.28 | 0.81 | −1.61 to 3.63 |
| Connectedness with Nature | 0.22 | 0.61 | −2.01 to 2.01 |
| Self-Esteem | 0.83 | 0.76 | −1.70 to 3.34 |

Note: Based on an analytic sample of 253.

### 6.2. Principles of Balanced Identity Theory

Zero-order correlations were used to examine the strength and direction of relationships between the concepts. There was a significant correlation between explicit attitude toward nature for both connectedness with nature and self-esteem, respectively, $r = 0.15$, $p < 0.05$; $r = 0.16$, $p < 0.01$. No other correlations between explicit measures were significant. Implicitly, there were significant correlations between the three IATs. The attitude toward nature IAT was positively related to both connectedness with nature ($r = 0.18$, $p < 0.01$) and self-esteem, $r = 0.15$, $p < 0.05$. The connectedness with nature IAT was also significantly related to self-esteem IAT, $r = 0.19$, $p < 0.01$.

To test our primary hypotheses, we used hierarchical regressions to examine the predictive nature of the explicit and implicit relationships separately, following the 4-test sequence of balanced identity. The analysis uses each of the legs of the triadic identity model as the criterion variable, as predicted by the multiplicative value of the other two legs [47]. If the identity is balanced, this multiplicative term should be statistically significant (test *a*). Then on step two of the hierarchical regression, the two main effects are entered into the regression equation. If the identity is balanced, the multiplicative term should remain positive (test *b*), the change in variance explained should not increase significantly (test *c*), and the two individual terms should not contribute significantly to the equation (test *d*). These analyses are conducted three times, once for each leg of the triadic identity. Note that this analytic procedure deviates from the typical moderated regression analysis in which the multiplicative term is added last.

#### 6.2.1. Predicting Attitudes toward Nature

The predicted patterns were found for implicit attitudes toward nature, but not for explicit attitudes toward nature measures. Figure 2 highlights these findings. For the implicit measures, when the multiplicative term was entered into the regression on step 1, the equation was statistically significant ($R = 0.20$, $R^2 = 0.04$, $F(1, 251) = 10.83$, $p < 0.01$), and the regression coefficient was positive, $b = 0.24$, $\beta = 0.20$, $t = 3.29$, $p < 0.01$. The multiplicative term was the product of the IAT measures for self-esteem and connectedness with nature, and this term was used to predict attitudes-toward-nature IAT scores.

In the second step of the hierarchical regression, self-esteem and connectedness with nature were each added as individual predictor variables. According to balanced identity theory, if the identity is balanced, the associated *b*-weight for the multiplicative term should remain positive at step 2, the overall model should not show a statistically significant increase in the percentage of variance explained, and the two predictor variables should not reach statistical significance [47]. These hypothesized patterns were found for the implicit measures. When the predictor variables were entered along with the multiplicative term, the percentage of variance explained did not increase significantly ($F(2, 249) = 0.81$, $p = 0.45$), the multiplicative term remained positive ($b = 0.02$), and the two individual coefficients for connectedness with nature ($b = 0.17$, $\beta = 0.15$, $t = 1.23$, $p = 0.22$), and self-esteem ($b = 0.12$, $\beta = 0.11$, $t = 1.04$, $p = 0.30$) were not statistically significant.

For the explicit measures, the multiplicative effect entered in step 1 was statistically significant ($R = 0.19$, $R^2 = 0.04$, $F(1, 251) = 8.99$, $p < 0.01$), with a significant multiplicative term, $b = 0.21$, $\beta = 0.19$, $t = 3.00$, $p < 0.01$. When the predictor variables were added to the equation in step 2, the change in variance explained was not significant, $F(2, 249) = 2.18$, $p = 0.12$. The multiplicative term remained positive ($b = 0.13$), and the connectedness with nature term was not statistically significant ($b = 0.08$, $\beta = 0.06$, $t = 0.62$, $p = 0.54$). However, there was a significant effect for self-esteem, $b = 0.14$, $\beta = 0.13$, $t = 2.08$, $p < 0.05$.

**Explicit Measures**

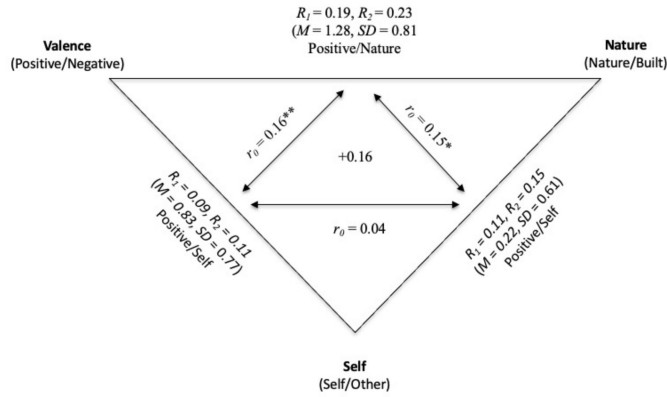

Average $R^2$ increments, Step 1: < 1%, Step 2: 2%

**Implicit Measures**

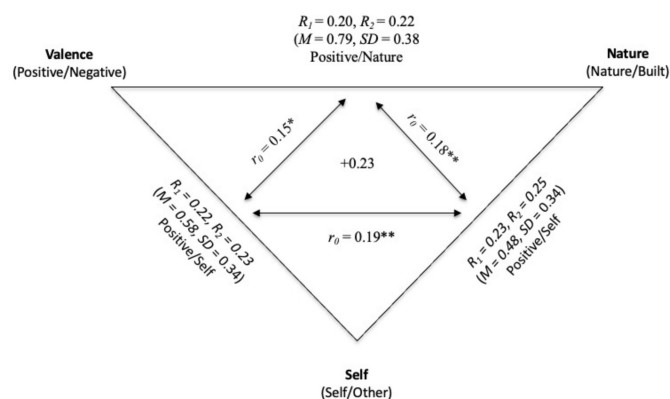

Average $R^2$ increments, Step 1: 5%, Step 2: 5%

**Figure 2.** Hierarchical regression analyses for the triadic structure of explicit and implicit environmental identity. Note: * = significant at $p < 0.05$; ** = significant at $p < 0.01$.

6.2.2. Predicting Connectedness with Nature

The same analyses were conducted using the two remaining variables as the criterion (connectedness with nature and self-esteem). These analyses were conducted to obtain the full tripartite relationship between these variables when looking at environment identity. The above predict patterns were found for implicit attitude toward nature, but not explicit attitude toward nature measures (see Figure 2).

For the implicit measures, the interaction effect entered in step 1 was significant, $F(1, 251) = 14.13$, $p < 0.001$, $b = 0.19$, $\beta = 0.23$, $t = 3.76$, $p < 0.001$. In step 2, the multiplicative term should remain positive, with the $R$-change not reaching statistical significance. In addition, when entered into the model separately, neither of the predictor variables should reach statistical significance. This was partially found for the implicit measures, but not for the explicit measures. For the implicit measures, when the predictor variables were entered with the multiplicative term, the change in variance explained was not significant, $F(2, 249) = 1.20$, $p = 0.30$. However, the multiplicative term was no longer positive, $b = -0.06$. Neither attitudes toward nature nor the self-esteem term were significant, respectively, $b = 0.18$, $\beta = 0.20$, $t = 1.49$, $p = 0.14$; $b = 0.15$, $\beta = 0.22$, $t = 1.34$, $p = 0.15$.

For the explicit measures, the interaction effect entered in step 1 accounted for 1% of the variance ($R = 0.11$, $R^2 = 0.01$) and was not significant, $F(1, 251) = 3.11$, $p = 0.08$, $b = 0.04$, $\beta = 0.11$, $t = 1.76$, $p = 0.08$. When the predictor variables were entered along with the multiplicative term in step 2, the change in variance explained was not statistically significant, $F(2, 249) = 1.40$, $p = 0.25$. The multiplicative term remained positive ($b = 0.04$), and neither the self-esteem term ($b = -0.04$, $\beta = -0.06$, $t = -0.40$, $p = 0.69$) nor the attitude toward nature term ($b = 0.08$, $\beta = 0.10$, $t = 1.04$, $p = 0.30$) were significant.

### 6.2.3. Predicting Self-Esteem

Finally, these relationships were examined using self-esteem as the criterion. The above predicted patterns were found for attitudes toward nature at the implicit level but using the explicit measures (see Figure 2). For the implicit measures, the interaction effect entered in step 1 accounted for 5% of the variance ($R = 0.22$, $R^2 = 0.05$) and was statistically significant, $F(1, 251) = 13.06$, $p < 0.001$, $b = 0.20$, $\beta = 0.22$, $t = 3.61$, $p < 0.001$.

In step 2, the associated $b$-weight for the multiplicative term should remain positive, with the $R$ not reaching statistical significance. In addition, when entered into the model separately, both of the predictor variables should not reach statistical significance. Results showed that when the predictor variables were entered along with the multiplicative term, the change in variance explained was not statistically significant, ($F(2, 249) = 0.30$, $p = 0.74$) and the multiplicative term remained positive ($b = 0.08$). The attitude toward nature term ($b = 0.07$, $\beta = 0.08$, $t = 0.68$, $p = 0.50$), and the connectedness with nature ($b = 0.11$, $\beta = 0.15$, $t = 0.75$, $p = 0.46$) were not statistically significant.

For the explicit measures, the interaction effect entered in step 1 accounted for 1% of the variance ($R = 0.09$, $R^2 = 0.01$) and was not statically significant, $F(2, 251) = 2.16$, $p = 0.14$, $b = 0.07$, $\beta = 0.09$, $t = 1.47$, $p = 0.14$. When the predictor variables were entered along with the multiplicative term in step 2, the percentage of variance explained increase significantly, $F(2, 249) = 2.95$, $p = 0.05$. The multiplicative term remained positive ($b = 0.11$) and the connectedness with nature term was not significant, $b = -0.13$, $\beta = -0.10$, $t = -0.79$, $p = 0.43$. However, there was a significant effect of the attitude toward nature term, $b = 0.13$, $\beta = 0.14$, $t = 2.12$, $p < 0.05$.

## 7. Discussion

In this present research, the social psychological concepts of attitudes toward nature, connectedness with nature, and self-esteem were integrated to create a triadic structure of environmental identity. Using balanced identity theory as a guiding framework [47], we tested the balanced-congruity principle for environmental identity. Results showed the expected pattern of relationships, suggesting a tendency for individuals to establish balanced identities with regard to self, nature, and attitude. In essence, when an individual scores highly on one of these, it is likely that they will score highly on the other two.

The reported analyses followed the sequence outlined by Greenwald et al. (2002) [47]. The results from a series of hierarchical regression analyses showed the hypothesized multiplicative effect for each of the IAT measures: self, attitude, and self-esteem. In addition, there was a positive association between the multiplicative term and the criterion (whether it be attitude toward nature, connectedness with nature, or self-esteem), and when the individual predictor variables were entered into the equation, the variance explained did not increase. That is, the implicit measure data were largely consistent with the balance-congruity principle from balanced identity theory. The findings suggest that individuals hold balanced identities, such that attitudes toward nature, connectedness, and self-esteem exist in a balanced triadic structure.

The current findings add to the growing body of literature that support balanced identity theory [59,64–71]. These studies have shown the tendency for individuals to maintain balanced identities across a range of domains, such as gender and racial identity. In addition, studies of balanced identity theory have typically found stronger results for balance using implicit than explicit measures. In a meta-analysis of 14 studies testing

balanced identity theory, implicit measures were found to produce larger effect sizes than the parallel explicit measures [59]. In addition, the implicit measures showed a substantially better fit for the balanced-congruity principle of the balanced identity theory than explicit measures [59,60]. The results reported in the current paper are in line with these prior studies.

### 7.1. The Difference in Balance between Explicit and Implicit Measures

Greenwald et al. (2002) proposed two reasons why the findings using explicit measures were not consistent with the results using implicit measures in the balanced congruity hypothesis [47]. First, self-report measures may be unable to tap into the strength of the association between concepts due to the introspective limits of the participant. Secondly, the participant may be utilizing response factors, such as demand characteristics and evaluation apprehension tactics, in their explicit responses. Both of these factors were not present within implicit measures. With this in mind, it is possible for different balanced identity structures to occur for explicit and implicit measures because explicit connectedness with nature may include (as proposed by Greenwald et al., 2002) [47] self-presentation methods, demand characteristics, social normative influence, and culture in its measurement. Both explicit and implicit evaluations measure connectedness with nature, but the unique aspects each brings creates differences among these balanced structures [72–75]. This overlap and uniqueness between explicit and implicit measures have been seen with connectedness with nature as well [76–79].

Explicit and implicit evaluations may also be measuring different aspects of connectedness with nature itself (e.g., state versus trait connectedness with nature) [54]. The IAT is typically used to measure stable associations rather than temporary states of mind [80]. Perhaps this accounts for the differences in the balanced congruity hypothesis between explicit and implicit measures. In addition, the experiences that connect people with nature may produce discrepancies between explicit and implicit balanced environmental identities. For instance, experiences in nature that build connectedness with nature are often emotional and experiential (e.g., experiences of awe) instead of logical and rational [81,82]. As such, these experiences tend to be nonverbal (compared to deliberate, verbal reflections) and may simply be more attuned to association-based knowledge [74,83]. These experiences that are more attuned to association-based knowledge could explain why the implicit measures (which capture the association-based relations) show more substantial balance effects than the explicit measures. Past research on the IAT has found that the implicit measures capture nonverbal aspects of experiences and stimuli such as physical attractiveness [75] or subliminal information [84] more so than language-based knowledge. Taken together, past research on the IAT and differences between explicit and implicit measures might explain why the current outcomes for balanced environmental identity were obtained for implicit measures but not for explicit measures. Future research should continue to explore the differences between explicit and implicit measures when examining a balanced environmental identity.

### 7.2. Self-Esteem and Environmental Identity

One of the exciting findings from this work is the novel integration of self-esteem into environmental identity. That connectedness and attitudes toward nature tend to follow the balance-congruity principle is not particularly surprising, and it has been proposed in previous studies [8,43,85–87]. But the importance of self-esteem in environmental identity is relatively new. Prior research has shown the mood-elevating effects of nature experiences [88], and the association between spending time in nature and positive mental health [27,89]. In addition, research has shown that connectedness with nature correlates with measures of well-being, including both hedonic well-being (feeling good) and with eudaimonic well-being (functioning well) [90]. But prior work has not directly linked self-esteem with attitudes toward nature, or connectedness with nature. The application of

balanced identity theory to environmental identity has the potential to generate new lines of inquiry, and new hypotheses about the role of self-esteem in efforts to promote sustainability.

While not tested directly in prior research, the role of self-esteem in environmental identity has roots in several other research traditions. For example, studies of interpersonal relationships have shown a link between social connectedness and self-esteem [91–93]. In the environmental literature, research has shown that spending time in nature can increase both connectedness with nature (as summarized below) *and* self-esteem (in addition to the mood-enhancing effects cited above) [94]. It feels good to be in nature, and we feel better about ourselves after nature experiences. As Clayton (2003) speculated in her early work on environmental identity:

By allowing people the time and space to think about their own values, goals, and priorities, as well as, perhaps, providing relief from the usual concerns of self-presentation, the natural environment can play a vital role in the extent to which we define ourselves to ourselves . . . The natural environment may enable a person to become both a more perceptive knower and a more positively valued known. We understand ourselves better and like ourselves more [43] (p. 49).

### 7.3. Practical Applications to Promote Sustainability

The findings reported in this paper also have practical applications for efforts to promote sustainability. Prior research has shown that certain types of nature experiences can increase connectedness with nature. For instance, spending time in a zoo-like setting [20,21], engagement campaigns [95], environmental education [23,96], and creating nature art [9] have all been found to increase connectedness with nature. The current findings suggest that experiences that increase connectedness, should exert pressure on the related concepts of attitudes toward nature, and self-esteem. That is, individuals who show an enhanced level of connectedness with nature following a nature experience should show commensurate changes in their attitudes about nature and feel better about themselves. However, given the findings that identity balance tends to occur more prominently at the implicit level, we might expect that these changes would not be immediate, nor lasting. Instead, the changes will likely require repeated experiences, which overtime will reconcile through the person's cognitive schemas to establish a balanced environmental identity.

Environmental identity researchers have speculated that environmental identity may be a necessary component in developing a relationship with the natural world and, in turn, can foster pro-environmental behaviors. This predicted relationship between environmental identity and pro-environmental behaviors has been shown in prior studies [43,97–99]. Thus, an environmental identity can lead to developing care for the environment and, in turn, acting in a sustainable way. As increasing connectedness with nature promotes a more balanced environmental identity, this balanced environmental identity then should foster sustainable behaviors. Again, this relationship between connectedness with nature and pro-environmental behaviors has also been found in prior research [88,100,101]. Implications from the current study then suggest that by creating a more balanced environmental identity through increasing connectedness with nature (e.g., by spending time in nature), pro-environmental behaviors should follow. This has important application for those interested in promoting sustainable behavior. Programs geared toward increasing sustainability may want to focus on increasing connectedness with nature, environmental attitudes, or self-esteem. This, in turn, should balance one's environmental identity and then people should be more likely to act in sustainable ways. However, the current study does not directly connect balanced environmental identity with behavior. Thus, the question about the integration of behavior into the balanced identity theory framework remains for future research.

### 7.4. Limitations

While balanced identity theory offers an exciting integration of social psychological constructs for understanding environmental identity, there are several notable limitations in

the work presented in this paper. First, within the balanced identity theory framework there is the lack of individual-level metric. Balanced identity theory is tested at the aggregate level, and the results suggest a tendency for individuals to maintain balance in their levels of connectedness, attitudes, and self-esteem. But research using balanced identity theory has not offered a procedure for measuring the strength of a person's identity. This also remains for future research.

Another limitation in this study was that data was collected from mainly female undergraduate college students in California. Prior research has suggested that identities are personally subjective and socially and culturally structured. People experience nature not only by themselves, but with others, and that these social relationships play an important role in developing an environmental identity [102]. They also suggest that "our experience of nature in the company of others affects our understanding of what nature signifies as well as the way we conceptualize our own relationship with nature" [102] (p. 66). In a study on the role of gender, race, and ethnicity in undergraduate student environmental identity development, gender stereotypes affected the development of undergraduate environmental identity [103]. Those around us influence these experiences and meaning can be taken by watching the actions and behaviors of others. Zavestoski (2003) drew similar conclusions but added that to fully develop and express environmental identity, the social environment must nurture and affirm one's environmental identity [104]. Thus, the social and cultural context of one's environmental identity must be taken into account.

In addition, past research has pointed toward identities as multifaceted and complex [105]. Identity types are organized hierarchically in one's self-concept, with those identities toward the top of the hierarchy having more influence toward self-congruence and therefore are more self-defining. In a study on creating sustainable identities. An affluent identity was found to be higher in the hierarchy than holding an environmental identity, and holding the affluent identity resulted in less sustainable behavior (higher energy consumption) [106]. Thus, the types of identities one holds are important for understanding sustainability. In this study, the only identity that was examined was an environmental identity.

Finally, in the current study, the triadic structure of an environmental identity was examined, using the concepts attitudes toward nature, connectedness with nature, and self-esteem. Other constructs related to pro-environmental behaviors (e.g., affective concern for nature) may also be part of one's environmental identity or other types of identities (e.g., personal identity, social identity) may influence sustainability. Future research is needed to examine these balanced environmental identity structures outside of the college population, including additional populations (e.g., genders and ages) and in various locations. In addition, future research needs to examine environmental identity in light of other constructs and/or identities related to sustainability.

In conclusion, balanced identity theory provides a novel framework for understanding the structure of environmental identity. Using this theory as a guide, concepts of attitudes toward nature, self-esteem, and connectedness with nature were integrated to form the triadic structure of environmental identity. In essence, when an individual scores highly on one of these, it is likely that they will score highly on the other two. This finding has important implications for practitioners interested in fostering environmental identities and promoting sustainability.

**Author Contributions:** Conceptualization, C.M.B. and P.W.S.; methodology, C.M.B.; formal analysis, C.M.B.; investigation, C.M.B.; writing—original draft preparation, A. W., C.M.B. and P.W.S.; writing—review and editing, A.W., C.M.B. and P.W.S.; All authors have read and agreed to the published version of the manuscript.

**Funding:** This research received no external funding.

**Institutional Review Board Statement:** The study was conducted according to the guidelines of the Declaration of Helsinki, and approved by the Institutional Review Board (or Ethics Committee) of Claremont Graduate University (protocol code 2011-023 and date of approval: 03/21/2011).

**Informed Consent Statement:** Informed consent was obtained from all subjects involved in the study.

**Data Availability Statement:** Data supporting reported results can be found at: Bruni, C.M. *Applying balanced identity theory to environmentalism.* PsycEXTRA Dataset [Internet]. American Psychological Association (APA), 2012. https://doi.org/10.1037/e521512014-142 (accessed on 5 June 2021).

**Conflicts of Interest:** The authors declare no conflict of interest.

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
