# Peer review of "The Balanced Structure of Environmental Identity"

_sustainability, doi:10.3390/su13158168_

Round 1

Reviewer 1 Report

This study is significant for practitioners and researchers alike in the area of environmental studies and environmental psychology. The findings extend our understanding of environmental identity. The implications of the study should be extended to show how environmental identity can be promoted and long-term sustainable behaviour maintained. Citing more varied research on environmental identity might not only support the literature review section, but also enhance the implications of the study section. I recommend including connections to environmental attitudes formation (see e.g. Domalewska, D. (2021) “A longitudinal analysis of the creation of environmental identity and attitudes towards energy sustainability using the framework of Identity Theory and big data analysis” Energies 14(3), https://doi.org/10.3390/en14030647 or Hurth, V. “Creating sustainable identities: The significance of the financially affluent self”, Sustainable Development 18, https://doi.org/10.1002/sd.453.

The last sentence of the paper is unfinished.

The abstract has been well-structured; however, it should more clearly explain what the major findings of the study were.

Author Response

Response:

Thank you for your review of our paper.  To address your comments/suggestions we’ve made the following changes:

  1. The implications of the study should be extended to show how environmental identity can be promoted and long-term sustainable behaviour maintained.

Response: Thank you for this suggestion, and for the positive comments about our manuscript. Following your suggestion, we’ve broadened the implications section to discuss how environmental identity can be promoted and the potential connection between identity and behavior.

  1. Citing more varied research on environmental identity might not only support the literature review section, but also enhance the implications of the study section. I recommend including connections to environmental attitudes formation (see e.g. Domalewska, D. (2021) “A longitudinal analysis of the creation of environmental identity and attitudes towards energy sustainability using the framework of Identity Theory and big data analysis” Energies 14(3), https://doi.org/10.3390/en14030647 or Hurth, V. “Creating sustainable identities: The significance of the financially affluent self”, Sustainable Development 18, https://doi.org/10.1002/sd.453.

Response: Thank you for this suggestion, and for the two relevant citations. We have added both of these articles to the introduction, along with a brief description of each and their relevance to the current work. We’ve also added some additional articles on environmental identity, both in the literature review and discussion sections.  In addition, we added a section to the introduction on environment attitudes and their connection to connectedness with nature, pro-environmental behaviors, and psychological theory.

Reviewer 2 Report

- Abstract section should refer to the study findings, methodologies, discussion as well as conclusion. It is suggested to present the abstract in 200-250 words paragraph.

- Methods section determines the results. Kindly focus on three basic elements of the methods section.
a. How the study was designed?
b. How the study was carried out?
c. How the data were analyzed?

- The subject addressed is within the scope of the journal.

- However, the manuscript, in its present form, contains several weaknesses. Appropriate revisions to the following points should be undertaken in order to justify recommendations for publication.

- More suitable title should be selected for the article.

- It is suggested to add articles entitled “Izadpanah et al. Evaluation of the Architectural Features and Physical Environment in Early Childhood Education Framework”, “Modranský et al. Innovation Management and Barriers – Creating Space for Innovation and Organizational Change” and “Okeke et al. City as Habitat; Assembling the Fragile City” to the literature review.

- For readers to quickly catch your contribution, it would be better to highlight major difficulties and challenges, and your original achievements to overcome them, in a clearer way in the abstract and introduction.

- Some key parameters are not mentioned. The rationale for the choice of the particular set of parameters should be explained in more detail. Have the authors experimented with other sets of values? What are the sensitivities of these parameters on the results?

- The captions of Figures 1 and 2 are missing in the manuscripts.

Page 5, Lines 230; This paragraph is not clear: The thermometer measure followed the procedure outlined above. Participants rated the extent to which they viewed themselves and other people to be more similar to built environments or nature, using a thermometer scale with anchors of 0(built) on the left, 5 (neutral) in the middle, and 10 (nature) on the right. 

- Some assumptions are stated in various sections. Justifications should be provided on these assumptions. Evaluation on how they will affect the results should be made.

- “Notation” should be added to the article.

- DOI of the references must be added (you can use “https://crossref.org/").

Author Response

Thank you for these suggestions. To address your comments/suggestions we’ve made the following changes:

  • Abstract section should refer to the study findings, methodologies, discussion as well as conclusion. It is suggested to present the abstract in 200-250 words paragraph.

Response: We have reviewed the guidelines for Frontiers with regard to an abstract, and worked to provide a clear and compliant summary. Our revised abstract is 179 words, and Frontiers guidelines specifies a maximum of 200 words.  We added additional information about the study findings, methodologies, discussion and conclusion to the abstract.

  1. Methods section determines the results. Kindly focus on three basic elements of the methods section.
    How the study was designed?
    b. How the study was carried out?
    c. How the data were analyzed?
  2. Some key parameters are not mentioned. The rationale for the choice of the particular set of parameters should be explained in more detail. Have the authors experimented with other sets of values? What are the sensitivities of these parameters on the results?

Response: With regard to methods and parameters we used, we have worked to streamline the description of the design, implementation, and analysis. Our goal is to provide a readable and accurate description. Hopefully the edits that we have made will improve the clarity.

  1. More suitable title should be selected for the article.

Response: We did not change the title of our paper as we feel it adequately highlights the topic being discussed in this manuscript, namely balanced identity theory, and environmental identity. We wanted a title that was short and included the concept of “balance” and “environmental identity.” We are open to alternative titles, but this seemed to capture the essence of the paper.

  1. It is suggested to add articles entitled “Izadpanah et al. Evaluation of the Architectural Features and Physical Environment in Early Childhood Education Framework”, “Modranský et al. Innovation Management and Barriers – Creating Space for Innovation and Organizational Change” and “Okekeet al. City as Habitat; Assembling the Fragile City” to the literature review.

Response: Thank you for the recommended articles.  We’ve considered each of these citations, and while they provide important findings, we felt that they were outside the scope of our current article.  However, we did recognize the need for additional literature and have added some additional articles on environmental identity, both in the literature review and discussion sections.  In addition, we’ve added a new section to the introduction on environment attitudes and their relationship with connectedness with nature, pro-environmental behaviors, and relevant psychological theory.

  1. For readers to quickly catch your contribution, it would be better to highlight major difficulties and challenges, and your original achievements to overcome them, in a clearer way in the abstract and introduction.

Response: Thank you for this suggestion. We have edited the abstract to include additional clarification of the key contributions of our research findings.  In addition, we’ve added additional limitations of the study to our discussion section, and along with suggestions for future research to overcome them.

  1. The captions of Figures 1 and 2 are missing in the manuscripts.

Response: Thank you for catching this error.  Figure captions for Figures 1 and 2 have been added and in-text notation of these figures has also been updated.

  1. Page 5, Lines 230; This paragraph is not clear: The thermometer measure followed the procedure outlined above. Participants rated the extent to which they viewed themselves and other people to be more similar to built environments or nature, using a thermometer scale with anchors of 0(built) on the left, 5 (neutral) in the middle, and 10 (nature) on the right. 

Response: Thank you for pointing this out. We have editing this paragraph in an effort to improve clarity.  The idea was that participants rated themselves on a sliding scale from built (0) to neutral (5) to nature (10), similar to the way they would rate themselves as connected with nature. These scales using 0 as a base anchor are important because they allow for direct comparisons to the implicit IAT scores, which center at zero.

  1. Some assumptions are stated in various sections. Justifications should be provided on these assumptions. Evaluation on how they will affect the results should be made.

Response: Thank you for this comment. We have provided additional justification of assumptions in an effort to provide clarity and avoiding confusing the reader.

  1. “Notation” should be added to the article.

Response: As an aid, we have used headers that follow the typical sections on: participants, materials, and procedure. Then we present an expanded Results section, divided into sections based on explicit and implicit measures. Given that we made separate sets of hypotheses about explicit and implicit measures, we hope that this structure will aid the reader in understanding our technical results.

  1. DOI of the references must be added (you can use “https://crossref.org/").

Response: Finally, doi’s have been added to the references.  Thank you for the helpful website.  It made finding these doi’s very easy.

Reviewer 3 Report

The reviewed manuscript explores an important aspect of environmental psychology. The research question is about the environmental identity linked to pro-environmental behavior. The study is sound both theoretically and empirically, although it needs some improvements and additions. My general advice is to communicate this study is a simpler form clear to sustainability experts from the research fields other than that of the authors.

  • Please, state your objective clearly in Introduction.
  • Methodology: can you provide the employed questionnaire?
  • Discussion is perfect, but research limitations need to be considered. Undergraduate students do not represent the entire society. Environmental identity may depend also on cultures and other national factors, and, thus, it may differ between countries. Again, your respondents are dominated by females, aren't they? – If so, this is a serious limitation. I also wonder about the age – these are chiefly young people, although I also wonder why these include people 50 years old (are these also undergraduate students?).
  • I suppose you need to consider more literature devoted to pro-environmental behavior – e.g., see papers by Evangelia Karasmanaki.
  • The manuscript needs section "Conclusions" listing the main findings in a simple and clear form. These findings should be clear even to non-specialists in environmental psychology. "Sustainability" is a broad-scope journal, and its readers are often experts in very different aspects of sustainability. I'm sure many of them will be interested in this study, but they will need easier (less "technical") explanations.
  • I think you need to explain the practical implications of the outcomes of your study, as well as to state their relevance to sustainability issues.
  • The manuscript needs better structuring. It is not always clear what is section and what is subsection.

Author Response

Thank you for these suggestions. Below is a point-by-point summary of our changes.

  • Please, state your objective clearly in Introduction.

Response: Thank you for this suggestion. We agree that the primary objectives and findings from the work should be provided early in the paper. In response, we have presented a statement of the objectives on page 4, the first sentence of the hypothesis section 

  1. Methodology: can you provide the employed questionnaire?

Response: We are unable to provide the employed questionnaire.  This questionnaire was given online.  This was clarified in the revision of the manuscript. In the spirit of open science, we are happy to provide screenshots of the online measures to interested readers, and we have provided our contact information.

  1. Discussion is perfect, but research limitations need to be considered. Undergraduate students do not represent the entire society. Environmental identity may depend also on cultures and other national factors, and, thus, it may differ between countries. Again, your respondents are dominated by females, aren't they? – If so, this is a serious limitation. I also wonder about the age – these are chiefly young people, although I also wonder why these include people 50 years old (are these also undergraduate students?).

Response: We agree and have added a limitations section that summarizes the limitations with our sample, known information about using undergraduate/female students, and culture.  We have also added a section on environmental identity to highlight your important point that, “environmental identity may depend also on cultures and other national factors, and thus, it may differ between countries.”

  1. I suppose you need to consider more literature devoted to pro-environmental behavior – e.g., see papers by Evangelia Karasmanaki.

Response: Thank you for the recommended citations.  We’ve reviewed these papers, but in the end, decided not to include them because they were outside the scope of our article.   Our focus is on “identity” and the question about the relationship between identity and pro-environmental behavior remains for future research. However, your comment prompted us to provide additional literature, and we have added some additional articles on environmental identity, both in the literature review and discussion sections.  In addition, we added a section to the introduction on environment attitudes and their connection to connectedness with nature, pro-environmental behaviors, and psychological theory.

  1. The manuscript needs section "Conclusions" listing the main findings in a simple and clear form. These findings should be clear even to non-specialists in environmental psychology. "Sustainability" is a broad-scope journal, and its readers are often experts in very different aspects of sustainability. I'm sure many of them will be interested in this study, but they will need easier (less "technical") explanations.

Response: We have added a Conclusions paragraph to the end of the discussion section, and clarified the key findings in both the abstract and discussion section of this paper.  With regard to our research methods and our results, we have worked to streamline the description of the design, implementation, and analysis. Our goal is to provide a readable and accurate description. Hopefully the edits that we have made will improve the clarity and make this manuscript more readable for those interested in sustainability more broadly

  1. I think you need to explain the practical implications of the outcomes of your study, as well as to state their relevance to sustainability issues.

Response: We have added practical implications of the outcomes of your study, specifically for maintaining and promoting pro-environmental behavior and sustainability more broadly.

  1. The manuscript needs better structuring. It is not always clear what is section and what is subsection.

Response: Thank you for this comment. As an aid, we have used headers that follow the typical sections on: participants, materials, and procedure. Then we present an expanded results section, divided into sections based on explicit and implicit measures. Given that we made separate sets of hypotheses about explicit and implicit measures, we hope that this structure will aid the reader in understanding our technical results.

Round 2

Reviewer 2 Report

The authors have successfully addressed all my concerns in the revised manuscript. Hence I recommend the acceptance of this paper.

Author Response

Thank you

Reviewer 3 Report

Dear Authors, I'm fully satisfied with your improvements. I just see that the standard formatting is not followed and the abstract is missed. Nonetheless, I think these are technical things, which can be fixed at the text rounds of the paper publishing.

Author Response

Thank you.  We have added back in the abstract, which was left out by the editorial team when converting our paper to their template.